# A new archosauromorph from South America provides insights on the early diversification of tanystropheids

**Tiane M. De-Oliveira**[1,2]*, **Felipe L. Pinheiro**[2], **Átila Augusto Stock Da-Rosa**[1,3],
**Sérgio Dias-Da-Silva**[1,3], **Leonardo Kerber**[1,4,5]

**1** Programa de Pós-Graduação em Biodiversidade Animal, Universidade Federal de Santa Maria (UFSM), Santa Maria, Rio Grande do Sul, Brazil, **2** Laboratório de Paleobiologia, Universidade Federal do Pampa, São Gabriel, Brazil, **3** Laboratório de Paleobiodiversidade Triássica, Departamento de Ecologia e Evolução, Universidade Federal de Santa Maria, Santa Maria, RS, Brazil, **4** Centro de Apoio à Pesquisa Paleontológica da Quarta Colônia (CAPPA), Universidade Federal de Santa Maria (UFSM), São João do Polêsine, Rio Grande do Sul, Brazil, **5** Museu Paraense Emílio Goeldi, Coordenação de Ciências da Terra e Ecologia, Belém, PA, Brazil

* tiane.m.deoliveira@gmail.com

**Data Availability Statement:** All relevant data are within the paper and its Supporting Information files.

## Abstract

After the Permo-Triassic mass extinction, the archosauromorph fossil record is comparatively abundant and ecologically diverse. Among early archosauromorphs, tanystropheids gained considerable attention due to the presence of extreme skeletal adaptations in response to sometimes overspecialized lifestyles. The origin and early radiation of Tanystropheidae, however, remains elusive. Here, a new Early Triassic archosauromorph is described and phylogenetically recovered as the sister-taxon of Tanystropheidae. The new specimen, considered a new genus and species, comprises a complete posterior limb articulated with pelvic elements. It was recovered from the Sanga do Cabral Formation (Sanga do Cabral Supersequence, Lower Triassic of the Paraná Basin, Southern Brazil), which has already yielded a typical Early Triassic vertebrate assemblage of temnospondyls, procolophonoids, and scarce archosauromorph remains. This new taxon provides insights on the early diversification of tanystropheids and represents further evidence for a premature wide geographical distribution of this clade. The morphology of the new specimen is consistent with a terrestrial lifestyle, suggesting that this condition was plesiomorphic for Tanystropheidae.

## Introduction

Archosauromorpha comprises an exceptionally diverse clade of diapsids, which originated during the Permian and progressively increased its diversity throughout the Mesozoic and Cenozoic eras. The most critical adaptive radiation of this clade took place following the aftermath of the Permian-Triassic mass extinction, resulting in a wide spectrum of occupation regarding both habitat and ecological niches [1,2]. After the Permian-Triassic crisis, the archosauromorph fossil record is considerably abundant and morphologically diverse, including

**Funding:** This study was financed by the Coordenação de Aperfeiçoamento de Pessoal de Nível Superior – Brasil (CAPES) – Finance Code 001. Fundação de Amparo à Pesquisa do Estado do Rio Grande do Sul (FAPERGS 16/2551-0000271-1 to FLP; 17/2551-0000816-2 to LK) and Conselho Nacional de Desenvolvimento Científico e Tecnológico (CNPq 407969/2016-0; 305758/2017-9 to FLP; 313494/2018-5 to AASR; 306352/2016-8 to SDS; 422568/2018-0; 309414/2019-9 to LK).

**Competing interests:** The authors have declared that no competing interests exist.

highly specialized herbivores (rhynchosaurs), large apex predators (erythrosuchids), aquatic predators (phytosaurs), armored crocodile-like forms (aetosaurs), and gracile dinosaur precursors [1–3].

One of the early archosauromorph clades that better illustrates the morphological disparity of the group is the Tanystropheidae, which comprises *Macrocnemus* Nopcsa 1930, *Tanystropheus* Wild, 1973, *Amotosaurus* Fraser and Rieppel, 2006, *Langobardisaurus* Renesto, 1994, and *Tanytrachelos* Olsen, 1979) [2,4–6]. Recently, *Boreopricea funerea* Tatarinov, 1978 and *Dinocephalosaurus orientalis* Li, 2003 were also recovered as phylogenetically closer to the tanystropheids than to other archosauromorphs [7]. Tanystropheidae is remarkable for including sometimes bizarre representatives with extreme morphologies [5]. Members of this clade are recognizable by their long necks, composed of eight (*Macrocnemus*) to thirteen (*Tanystropheus*) moderately to extremely elongated cervical vertebrae with very long and low neural spines [8,9]. Overall, the tanystropheid *bauplan* is regarded as evidence of a semiaquatic or even completely aquatic lifestyles [10–13]. However, recent studies failed to support a fully aquatic habit for tanystropheids, demonstrating that *Macrocnemus* was presumably terrestrial, whereas the lifestyle of the enigmatic *Tanystropheus*, the largest and most bizarre of all tanystropheids, remains enigmatic [9,14]. The fossil record of tanystropheids and related forms mostly come from the Middle/Late Triassic of Asia, Europe and North America [5,15], and the clade is exceptionally rare in Lower Triassic rocks (see [2,5,8,16,17]). Although the fossil record of Tanystropheidae was, until recently, restricted to the Northern Hemisphere, De-Oliveira *et al.* [18] described isolated cervical vertebrae that share synapomorphies with this clade from the Induan/Olenekian Sanga do Cabral Formation, which belongs to the Brazilian portion of the Sanga do Cabral Supersequence. In addition, a humeral fragment compatible with Tanystropheidae was recovered from Upper Permian strata of the Rio do Rasto Formation, Southern Brazil [19].

Based upon its tetrapod content, the Sanga do Cabral Formation is regarded as Lower Triassic, being correlated to the Katberg Formation of the South African Karoo Basin (*Lystrosaurus* Assemblage Zone) [20,21]. Although recent collection efforts substantially increased the number of archosauromorph specimens recovered from the Sanga do Cabral Formation (e.g. [18, 22, 23]), its diversity is still poor when compared to coeval deposits from South Africa, which have yielded the rhynchosaur *Noteosuchus* Broom, 1925, the well-known *Prolacerta* Parrington, 1935, and the archosauriform *Proterosuchus* Broom, 1903[24]. Nevertheless, the Sanga do Cabral Formation is one of the oldest Triassic sedimentary units yielding fossil vertebrates from South America and provides a unique opportunity to study the biotic recovery after the P/T boundary.

This contribution provides the description and phylogenetic analysis of a new archosauromorph species from the Sanga do Cabral Formation, which provides insights on the hidden western Gondwanan archosauromorph diversity after the Permo-Triassic global crisis, adding information on the early distribution and lifestyle of tanystropheid-like forms.

## Institutional abbreviations

AMNH, American Museum of Natural History, New York, USA; BP, Evolutionary Studies Institute, University of the Witwatersrand, South Africa; FMNH, Field Museum of Natural History, Chicago, Ilinois, USA; GR, Ruth Hall Museum of Paleontology, Ghost Ranch, Mexico; MCSN, Museo Civico di Storia Naturali Milano, Italy; MCSNB, Museo Civico di Scienze Naturali Enrico Caffi, Bergamo, Italy; MCZ, Museum of Comparative Zoology, Cambridge, USA; MSNM, Museo di Storia Naturale, Milano, Italy; NHMUK, Natural History Museum of the United Kingdom, London, UK; NMQR, National Museum Bloemfontein, Bloemfontein,

South Africa; PIMUZ, Paleontological Institute and Museum, Zürich, Switzerland; SMNS, Staatliches Museum für Naturkunde Stuttgart, Stuttgart, Germany; TMM, Texas Memorial Museum, Austin, Texas, USA; UFSM, Universidade Federal de Santa Maria, Santa Maria, Rio Grande do Sul, Brazil; UWBM, Burke Museum of Natural History and Culture, USA; UMCZ, University Museum of Zoology, Cambridge, UK; IVPPV, Institute of Vertebrate Paleontology and Paleoanthropology, Beijing; PIN, Palaentological Institute, Moscow.

## Methodology

### Material

The material, under collection number UFSM 11471, consists of an almost complete posterior limb with an articulated femur, tibia, fibula, and pes. The specimen also preserves portions of the pelvic girdle, sacral, and caudal vertebrae. The new specimen was collected at the locality Bica São Tomé, Sanga do Cabral Formation (Sanga do Cabral Supersequence, Paraná Basin) municipality of São Francisco de Assis, Rio Grande do Sul, Southern Brazil. It is housed at the paleontological collection of the Laboratório de Paleobiologia e Estratigrafia of the Universidade Federal de Santa Maria (UFSM), Santa Maria, Rio Grande do Sul, Brazil. No permits were required for the described study, which complied with all relevant regulations.

### Nomenclatural acts

The electronic edition of this article conforms to the requirements of the amended International Code of Zoological Nomenclature, and hence the new names contained herein are available under that Code from the electronic edition of this article. This published work and the nomenclatural acts it contains have been registered in ZooBank, the online registration system for the ICZN. The ZooBank LSIDs (Life Science Identifiers) can be resolved and the associated information viewed through any standard web browser by appending the LSID to the prefix "http://zoobank.org/". The LSID for this publication are: urn:lsid:zoobank.org:act:6C0BF61B-BD2C-42BD-BA44-47FC29D27DB2 and urn:lsid:zoobank.org:act:6BEA3D87--CA6F-444B-824E-E49E0FC12CA6. The electronic edition of this work was published in a journal with an ISSN, and has been archived and is available from the following digital repositories: PubMed Central, LOCKSS.

### Phylogenetic analyses

The phylogenetic analyses were carried out in order to access the affinities of UFSM 11471 with respect to other early archosauromorphs. UFSM 11471 was scored in the dataset of Pritchard *et al.* [6], as it includes a larger number of tanystropheids as terminal taxa when compared to other data matrixes. The dataset of Pritchard *et al.* [6] does not include *Jesairosaurus lehmani* Jalil, 1997, as an operational taxonomic unit (OTU). The phylogenetic position of this enigmatic archosauromorph, formerly regarded as a member of "Prolacertiformes", has not been tested in recent analyses until the work of Ezcurra [2]. This latter analysis recovered *J. lehmani* as the sister-taxon to the Tanystropheidae, even though it differs from tanystropheids in several important features. The close relationship between *J. lehmani* and the Tanystropheidae recovered by Ezcurra [2] led us to include this taxon as an OTU in a complementary explorative analysis. *J. lehmani* scoring followed the recognition of overlapping characters between the datasets of Pritchard *et al.* [6] and Ezcurra [2]. Additional scorable characters were taken from the literature. A similar procedure was applied in order to include *Dinocephalosaurus orientalis* as an additional OTU in this second survey. The analysis protocol of both independent experiments consisted of heuristic searches of 1000 replications using random

addition sequences followed by the Tree Bisection Reconnection (TBR) algorithm, retaining ten trees by replication (S1 Material).

## Systematic paleontology

DIAPSIDA OSBORN, 1903 (*SENSU* LAURIN 1991)

  ARCHOSAUROMORPHA HUENE, 1946 (*SENSU* GAUTHIER *ET AL.* 1988)

  *ELESSAURUS GONDWANOCCIDENS* GEN. ET SP. NOV. urn:lsid:zoobank.org: act:6C0BF61B-BD2C-42BD-BA44-47FC29D27DB2 urn:lsid:zoobank.org:act:6BEA3D87-- CA6F-444B-824E-E49E0FC12CA6

### Holotype

UFSM 11471 –A partially articulated hind limb associated with axial elements, composed of femur, tibia, fibula, pelvic girdle bones, sacral and caudal vertebrae, as well as an almost complete pes.

### Etymology

Genus named after *Elessar*, meaning 'elf-stone' in the fictional language Quenya, created by J. R. R. Tolkien. In Tolkien's Middle Earth universe, Elessar Telcontar is the name chosen by king Aragorn II, who, by his turn, is also known as Strider or 'longshanks'. The comparatively long zeugopodium of UFSM 11471 makes it a long-shanked animal, justifying the name. Termination -*saurus* from Greek, meaning 'lizard'. Species name derived from the supercontinent Gondwana and the Latin adjective *occidens*, 'from west', in a reference to the locality from where the new species was recovered.

### Diagnosis

*Elessaurus gondwanoccidens* differs from all other archosauromorphs based upon a unique combination of characters: second sacral vertebral rib elongated and distally bifurcated, with a robust articular surface; transverse processes of the caudal vertebrae inclined posterodorsally; strongly sigmoidal femur; tibia and fibula longer than femur; metatarsals increase in size from the first to the fourth toe; fifth metatarsal short, with a proximal hook-shaped end; presence of a calcaneal tuber.

### Locality and horizon

The specimen was collected at the locality known as Bica São Tomé, Sanga do Cabral Formation (Sanga do Cabral Supersequence, Paraná Basin), municipality of São Francisco de Assis, Rio Grande do Sul, Southern Brazil (29˚36′ 56″ S, 55˚03′ 10″ W) [22] (Fig 1). *Elessaurus gondwanoccidens* was collected in one of the five outcrops comprising the Bica São Tomé (outcrop 5 of Da-Rosa *et al.* [22]). An Induan-Olenekian age (251–247 Ma) [25] is inferred for this formation based on the presence of the parareptile *Procolophon trigoniceps* Owen, 1876, and comparisons with the *Lystrosaurus* Assemblage Zone of the South African Karoo Basin [21,22,24,26]. *Elessaurus gondwanoccidens* represents the most complete postcranial skeleton so far recovered from this unit, as Sanga do Cabral fossils are often fragmentary, with rare occurrences of associated elements.

## Description and comparison

*Elessaurus gondwanoccidens* holotype is composed of an almost complete hindlimb associated with pelvic girdle bones and partially articulated sacral and caudal vertebrae (Fig 2). Although

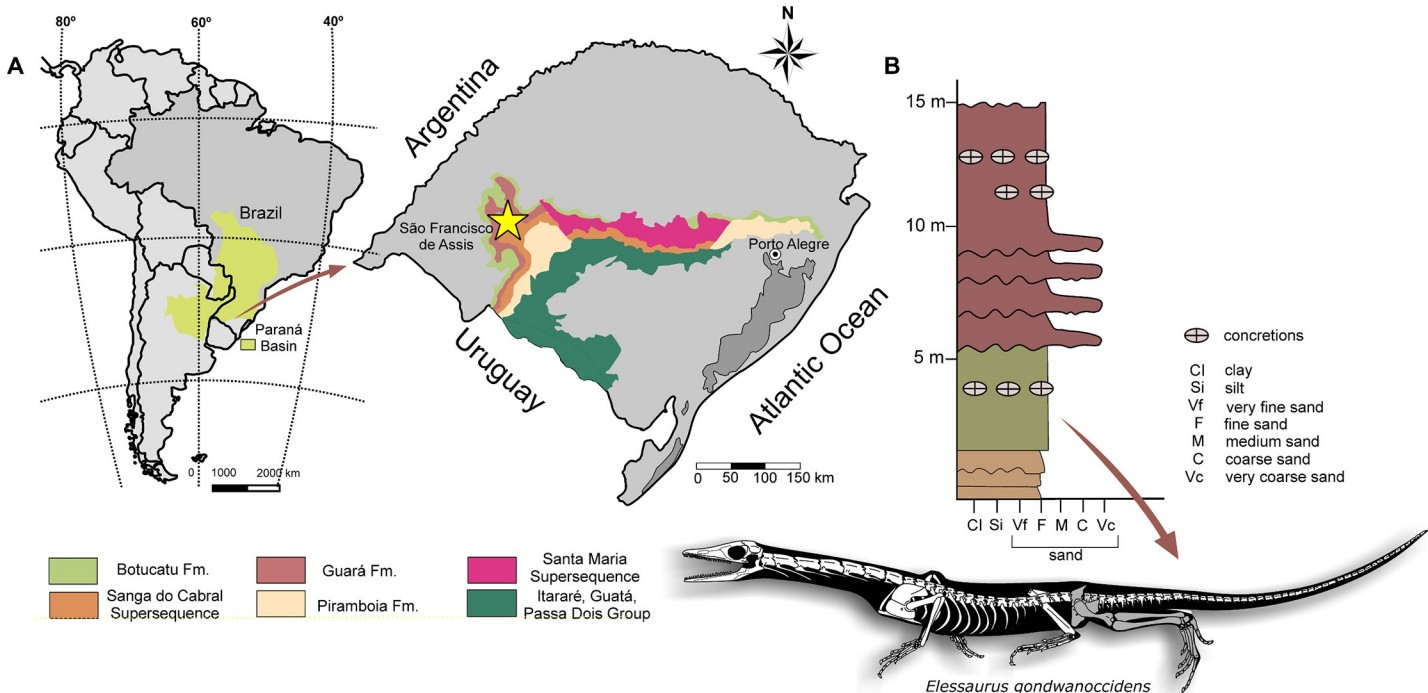

**Fig 1. Type locality of *Elessaurus gondwanoccidens* (UFSM 11471). A.** Geographic map evidencing the type-locality of *Elessaurus gondwanoccidens*, (São Francisco de Assis, Brazil); **B.** Simplified stratigraphic profile of the outcrop, showing the level where UFSM 11471 was found. Map was modified from Zerfass *et al.* [27] and stratigraphic profile modified from Da-Rosa *et al.* [22] and Pinheiro *et al.* [23]; silhouette adapted from Rieppel [15], showing bones preserved of UFSM 11471 in dark gray color. Reprinted from Da-Rosa *et al.* [22] under a CC BY license, with permission from Átila Stock Da-Rosa, original copyright 2009.

some elements show signs of compression (e.g. femur, tibia), all bones are close to a natural position, except for a slight displacement of some tarsal elements and distal phalanges missing in most digits. As will be discussed, the specimen is morphologically compatible with basal archosauromorphs, especially with Tanystropheidae.

## Vertebrae

Specimen UFSM 11471 preserves a complete second sacral vertebra associated with the first and second caudal elements (Fig 3). The first sacral comprises scattered fragments articulated to sacral II, which is well preserved and articulated to the ilium. The pleurapophysis of the second sacral vertebra is bifurcated distally, with a posterior process ending in a pointed tip. The distal end of the pleurapophysis is expanded and presents a wide triangular surface (as observed in dorsal view) which contacts the ilium.

The elongated and distally bifurcated pleurapophysis of the second sacral vertebra resembles the condition observed in tanystropheids such as *Macrocnemus* [13,28]. This condition, however, is also present in *Trilophosaurus buettneri* Case, 1928 and *Prolacerta*, as well as in some extant lizards [5,13,15,29]. *Augustaburiania vatagini* shows an alternative condition, where sacral vertebrae present small sacral ribs that deviate laterally in the middle of the centrum [8]. *Tanystropheus longobardicus*, the best-known tanystropheid, lacks distally bifurcated sacral ribs, whereas *Amotosaurus rotfeldensis* presents the bifurcation of the second sacral rib more anteroposteriorly expanded than *Elessaurus gondwanoccidens*, which presents the posterior process of the second sacral rib distally sharp, ending in a pointed tip, whereas in *Prolacerta* this process is terminally blunt [6]. *Tanytrachelos* does not present a bifurcated second

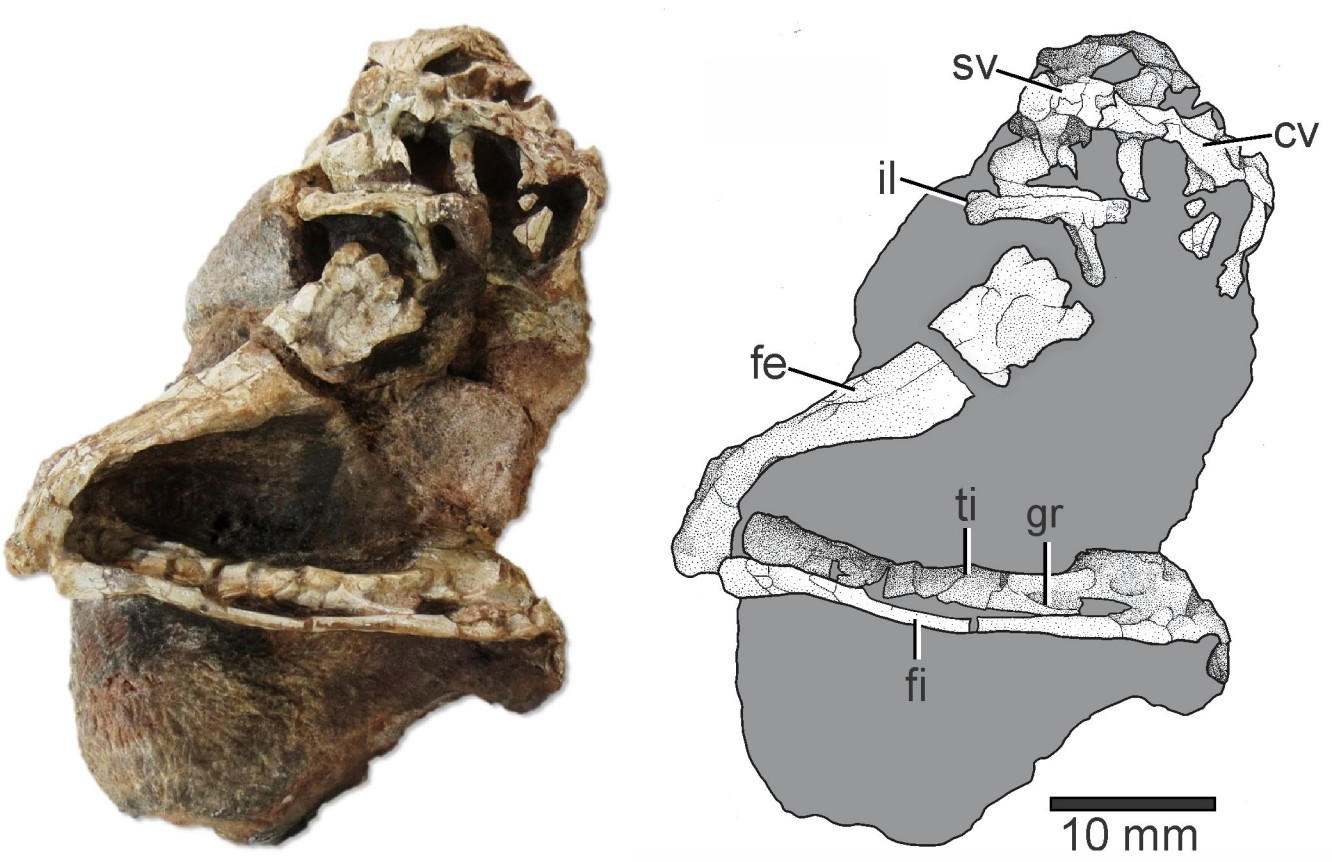

**Fig 2.** *Elessaurus gondwanoccidens* **(UFSM 11471) from the Sanga do Cabral Formation (Lower Triassic), Brazil.** Photograph and explanatory drawing respectively. Abbreviations: **fe**, femur; **ti**, tibia; **gr**, groove; **fi**, fibula; **il**, ilium; **sv**, sacral vertebra; **cv**, caudal vertebrae.

sacral rib, and, together with *Jesairosaurus lehmani*, it presents the transverse processes of the caudal vertebrae slightly inclined posterolaterally.

The anteriormost caudal vertebra has the same anteroposterior length of the second sacral and shows prominent transverse processes, projecting distinctly laterally to the pleurapophysis of the sacral vertebrae and the dorsal part of the ilium. In dorsal view, the transverse processes are posterolaterally directed. Although the transverse processes of the second caudal vertebra are scattered, they are slightly longer than those from the first caudal vertebra. The anterior caudal vertebrae also present distinct transverse processes in *Tanystropheus* and *Langobardisaurus* [10,30].

## Pelvic girdle

The pelvic girdle is fragmented, with its bones only partially preserved and exposed, being relatively small when compared to the large hindlimbs. The ilium is discernible in dorsolateral view, and it is not clear if both pubis and ischium are preserved. The ilium is expanded into a dorsal lamina that articulates with the large pleurapophysis of the sacral vertebra. Anteriorly to this, the ilium presents a strongly projected process. The dorsal margin of the iliac lamina is predominantly straight, with a horizontal orientation. The supracetabular surface is thickened, and the lateral surface of the acetabulum is roughly circular. The posterior process of the ilium is strongly developed, extending posteriorly to the acetabulum.

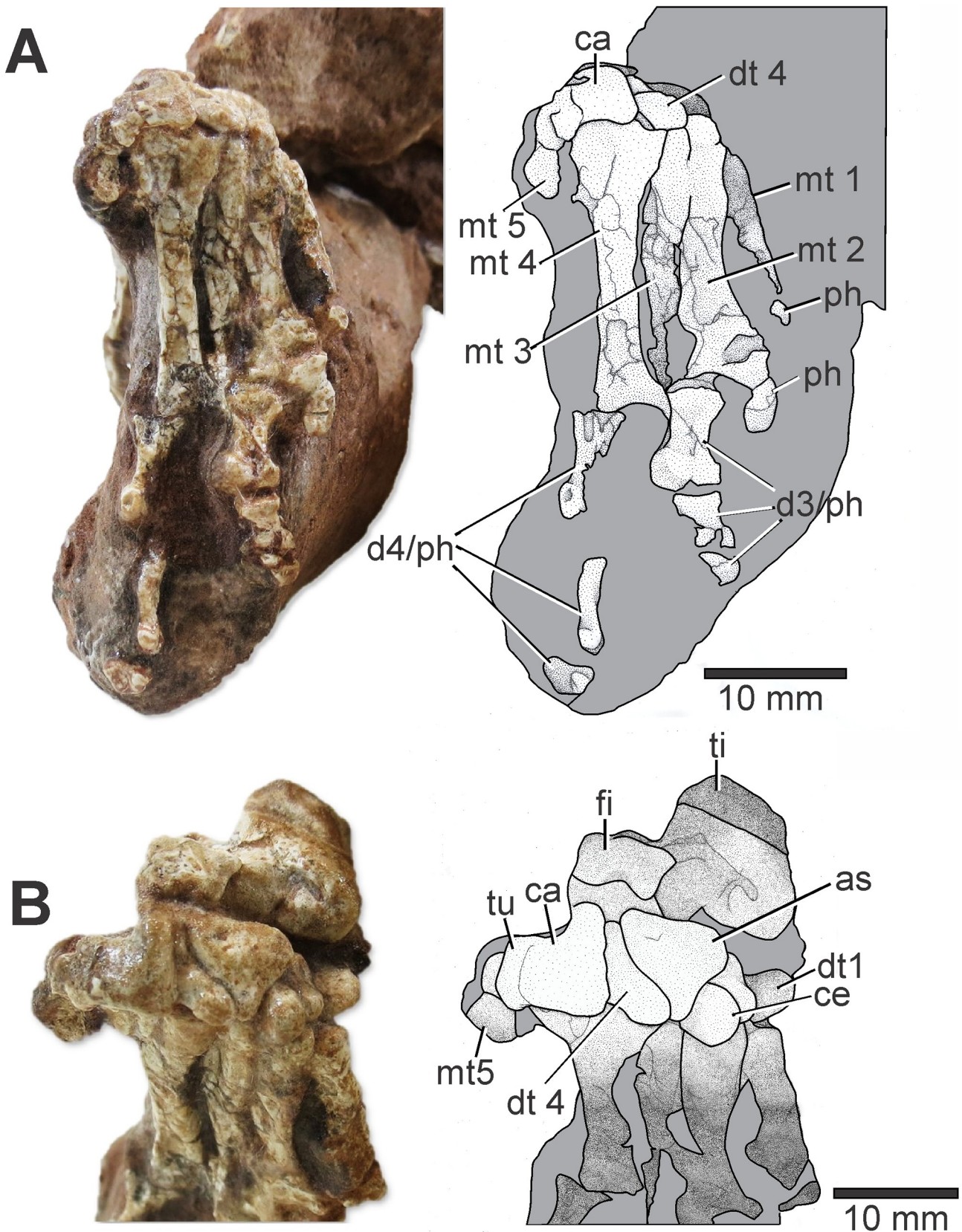

**Fig 3. Sacral and caudal vertebrae of *Elessaurus gondwanoccidens* (UFSM 11471) in dorsal view.** Photograph and explanatory drawing respectively. Abbreviations: sv2, second sacral vertebra; cv, caudal vertebrae 1–3.

The pelvic girdles of *Prolacerta* and tanystropheids are similar to that of *Elessaurus gondwanoccidens* regarding the presence of a posterodorsally directed triangular blade on the ilium. Although the iliac lamina of *Elessaurus* is partially broken, it is possible to discern a short preacetabular process similar to that observed in *Macrocnemus bassanii* (specimen T 2472). This is a robust process in this specimen, being also present (albeit less pronounced) in *Tanystropheus longobardicus* (specimen MSNM BES SC 1018) and *Dinocephalosaurus* (IVPP V13898), this latter presents the ilium with moderately developed preacetabular process and distinct dorsal iliac blade. Conversely, in *Prolacerta* the anterior margin of the ilium is convex, the same occurring in *Jesairosaurus lehmani* [13,15,16,31,32,33].

## Femur

With a total size of 64.85 mm, the femur is slightly shorter than the tibia and fibula. Its proximal end is poorly preserved and strongly compressed. The distal end of the femur is 16.55 mm in width, whereas the proximal part measures 15.44 mm. It is a gracile bone, with the ratio between the transversal width of the distal end and the total length of the bone being 3.91. Despite it shows signs of preservational compression, the femur is strongly sigmoid in lateral view, differing from the specimen GR-304 described by Pritchard *et al.* [5], which is nearly straight, turning distally from the surface of the proximal head.

Both *Tanystropheus longobardicus* (MSNM BES SC 265) and *Tanytrachelos* (YPM 7622) present a more subtle femur curvature when compared to *Elessaurus gondwanoccidens* [2,5,13,34,35], which resembles *Macrocnemus bassanii* (T 4355) and *Augustaburiania*, as both present gracile and strongly sigmoidal femora [8,15]. The femur of *Dinocephalosaurus* (IVPP V13898) is also a slightly curved element that resembles *Tanystropheus longobardicus* (MSNM BES SC 265) [33]. The shaft of the left femur of *Boreopricea funerea* (PIN 3708/1) also shows a slight sigmoid bend [36].

Probably due to poor preservation, the proximal surface of the femur has a quadrangular outline. The femoral head appears to be confluent with the shaft. The dorsolateral margin of the proximal portion of the femur is smooth and featureless, as is the transition between the femoral head and diaphysis. The femoral head is weakly expanded in tanystropheids (e.g., *Tanytrachelos ahynis*, AMNH FARB 7206, GR 301). In other basal archosauromorphs such as *Azendohsaurus madagaskarensis* Flynn *et al.* 2010 (UA 7-20-99-653), the proximal end is moderately expanded relative to the midshaft, as it is in some rhynchosaurs and early archosauriforms (e.g., *Proterosuchus alexanderi* Hoffman, 1965, NMQR 1484; *Erythrosuchus africanus* Gower, 1996 NHMUK 3592) [4]. The internal trochanter is present as a ridge-shaped process that defines a relatively wide intertrochanteric fossa, converging to the proximal end. The internal trochanter is continuous with the proximal articular surface. The transition from the plesiomorphic condition of a proximal trochanter, including an inner trochanter and a posterior trochanter (e.g. *Erythrosuchus africanus*; *Trilophosaurus buettneri*) to a large fourth trochanter and a larger trochanter (e.g. *Alligator*, Hutchinson, 2001; dinosaurs), occurs within archosauriforms [1,5,35]. Thus, the presence of an internal trochanter allied to the absence of a fourth trochanter strongly supports nesting of *Elessaurus gondwanoccidens* in a clade outside Eucrocopoda (defined by Ezcurra [2] as a suprageneric taxon including non-proterosuchian archosauriforms). A well-developed internal trochanter projecting from the proximal end of the femur is present in early archosauromorphs, including tanystropheids (e.g. *Macrocnemus bassanii* and *Tanystropheus longobardicus*). An internal trochanter that does not reach the

proximal surface of the femur is evident in rhynchosaurs and some archosauriforms, such as *Proterosuchus fergusi* Broom, 1903 and *Erythrosuchus africanus* [2,5,37,38].

Despite its compression, the femur of *Elessaurus gondwanoccidens* is slightly widened distally. Distally expanded femora (although in a lesser degree) occur in *Prolacerta broomi* (BP/1/2676), tanystropheids (e.g. *Tanystropheus*, Wild [10]; GR 301, Pritchard *et al.* [5]) and *Dinocephalosaurus*, that presents the proximal and distal ends of the femur distinctly expanded [33].

The distal end of the femur is marked by two delineated, unequal distal condyles, with the lateral larger than the medial one. They are distinctly expanded beyond the circumference of the femoral shaft. In distal view, the fibular condyle has a subtriangular lateral surface.

## Tibia and fibula

The poor preservation of the tibia hinders a proper morphological assessment of this bone. Tibia and fibula present both a length about 12% greater than the femur. The length relationship between tibia and fibula is considered an important phylogenetic feature in basal archosaurs (see Ezcurra [2]). Among those, the tibia is longer than the femur in basal pterosaurs (e.g. *Preondactylus* Wild, 1984), *Lagerpeton* Romer, 1971, *Dromomeron* Irmis *et al.* 2007, *Marasuchus* Sereno and Arcucci, 1994, *Pseudolagosuchus* Arcucci, 1987, basal ornithischians, *Eoraptor* Sereno *et al.* 1993 and most basal dinosauromorphs. [1]. Within non-archosauriforms, this characteristic is prominent among some tanystropheids (e.g. *Macrocnemus*). Compared to the forelimbs, the hindlimbs of *Macrocnemus* are strongly elongated [15], which is mainly acquired by an elongation of the tibia/fibula.

Besides, a short femur relative to the tibia and fibula is also present in *Prolacerta* (AMNH 9502, BP / 1/2676) [31]. Although also having a proportionally short femur, the *Prolacerta* specimen described by Spiekman [32] (UWBM 95529) shows a proportionally longer femur when compared to *Macrocnemus* and *Elessaurus*. Specimen UWBM 95529 has a femur measuring 72.4 mm, whereas the tibia is 69.3 mm in length. This is also the case of *Dinocephalosaurus* (IVPP V13898), which has a femur length of 116.2 mm and a tibia length of 63.7 mm. In this specimen, the fibula is longer than the tibia, but more delicately built and more distinctly curved than in *Elessaurus* [33].

Tibia and fibula at least 20% longer than the femur is one of the synapomorphies diagnosing *Macrocnemus* [5]. There are reports of proportional differences [39,40], among *Macrocnemus bassanii*, *Macrocnemus fuyuanensis*, and *Macrocnemus obristi*. Some authors even suggest that these differences may be related to sexual dimorphism [14]. The tibia of *Elessaurus gondwanoccidens* is much thicker than the fibula and has its distal end articulated with mesopodial elements. The distal end of the tibia is fragmented, which may be a result of pre-burial fracturing. Although the tibia is severely damaged, it is possible to observe a small groove in the lateral surface of its distal end (Fig 2), a feature only observed in some dinosaurs and proterochampsids (e.g. *Chanaresuchus* Reig, 1971, *Tropidosuchus* Arcucci, 1990) [1]. In the context of non-archosauriforms, thus, this may potentially be an autapomorphy of the new taxon described herein. However, this character might be also an artifact of poor preservation. The proximal end of the fibula is fragmented and compressed in proximal view, being rounded and symmetrical in lateral view. The area for insertion of the *M. iliofibularis* is evident by the presence of a distinct but low tubercle located near the proximal portion of the bone. The distal portion of the fibula is slightly asymmetrical in lateral view.

## Pes

Metatarsals and phalanges are articulated and arranged close to a natural position, with only a slight displacement of some elements. *Elessaurus gondwanoccidens* preserves the metatarsals

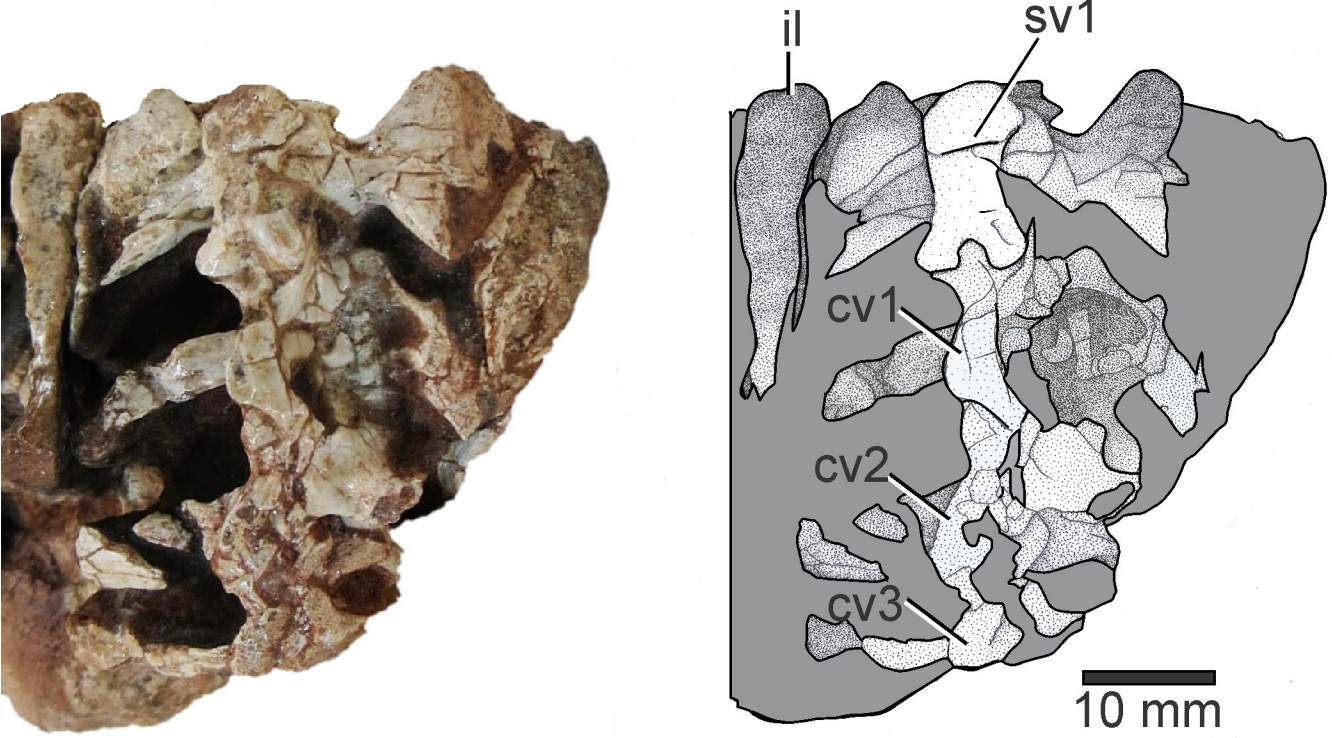

**Fig 4.** Plantar (A) and posteroplantar (B) views of the pes of *Elessaurus gondwanoccidens* (UFSM 11471) from the Sanga do Cabral Formation (Lower Triassic), Brazil. Photographs and explanatory drawings respectively. Abbreviations (**A**): **ca**, calcaneum; **dt 4**, distal tarsal 4; **mt**. metatarsal 1–5; **d3—d4**, digits; **ph**, phalange. (**B**) **ti**, tibia; **fi**, fíbula; **ca**, calcaneum; **as**. astragalus; **tu**, calcaneal tuber; **mt**, metatarsal 5; **dt 1**, distal tarsal 1; **ce**, centrale; **dt 3**, distal tarsal 3; **dt 4**, distal tarsal 4.

I-V, but phalanges of the fifth digit are missing. The metatarsals increase in length from the first to the fourth digit, where the metatarsal IV is distinctly larger than the III (Fig 4). The fourth digit of non-archosauriform archosauromorph pes (e.g., rhynchosaurs, *Trilophosaurus*, *Prolacerta*) is the longest, whereas digit 3 is the longest in *Euparkeria* Broom, 1913 (UMCZ T692) and all archosaurs in which this character can be accessed [1,2,41]. Metatarsals increase in size from the first to the fourth toe in *Macrocnemus bassanii* (T 2477; A III/208; T 2472), *Amotosaurus* (SMNS 54810), *Prolacerta*, and *Langobardisaurus* [15,17,30,42]. The metatarsus of *Tanystropheus* is asymmetrical, although not in the same degree as *Macrocnemus* and *Langobardisaurus* as, in *Tanystropheus*, the third metatarsal is the longest. *Tanytrachelos* (YPM 7540) apparently has a similar metatarsal configuration [13,34] as *Tanystropheus*.

The fifth metatarsal is short and has a proximal hook-shaped end: its proximal process is abruptly flexed and, as a result, the metatarsal is "L"-shaped in ventral view. This morphology is observed in *Macrocnemus bassanii*, allokotosaurs (e.g. *Pamelaria dolichotrachela* Sen, 2003, *Azendohsaurus madagaskarensis*), a few basal rhynchosaurs (e.g. *Noteosuchus colletti* Watson, 1912), *Boreopricea funerea* Tatarinov, 1978, *Prolacerta broomi*, and some archosauriforms (e.g. *Proterosuchus fergusi* Broom, 1903) [1,2]. Among tanystropheids, the morphology of metatarsal V of *Elessaurus gondwanoccidens* resembles the condition displayed by *Macrocnemus* (PIMUZ T AIII / 208) [15] and differs from that of *Langobardisaurus* and *Tanystropheus* (MSNM V 3730) [13], as these present a less pronounced hook-shaped element. In *Dinocephalosaurus* (IVPP V13898), all five metatarsals are preserved, the fourth being the longest in the

series and the first one being the shortest, and the fifth metatarsal is distinctly longer than the first and shows no trace of a 'hooked' shape [33].

The metatarsals diverge from the tarsus distally but overlap proximally. In digits I and II the distal phalanges are not preserved and the digits III and IV present two middle and one distal phalanges. The lack of some distal phalanges prevents an exact account of the phalangeal formula.

## Tarsals

Six tarsals are preserved in *Elessaurus gondwanoccidens*, including the proximal elements (astragalus and calcaneum), and four ossifications identified here as the distal tarsal elements I, III, IV, and the centrale (Fig 4B). Excepting the centrale, these elements are displaced laterally towards the calcaneum, distal to the tibia and proximal to the metatarsals I and II. A fifth distal tarsal is missing. From the distal elements, the fourth and the centrale are the largest. Distal tarsal IV is located between the astragalus and calcaneum, proximal to metatarsals III and IV, whereas distal tarsals III and I are placed medial to the centrale. Four distal tarsals occur in most early archosauromorphs (e.g., *Mesosuchus browni* Watson, 1912, SAM-PK 7416; *Protorosaurus speneri* Meyer, 1832; *Trilophosaurus buettneri*, TMM 31025–140) [4]. *Macrocnemus bassanii* presents four distal tarsals, one being the centrale. However, only three distal tarsals occur in *Macrocnemus fuyuanensis* and *Amotosaurus*, and only two in *Tanystropheus longobardicus* (MCSN BES SC 1018; MCSN V 3730) [4,15,43,44]. *Prolacerta broomi* [BP/1 2676] [31] was described as having a centrale in close contact with the mesial surface of the astragalus, besides four distal elements, of which the first three are small and fragmented. Colbert [42] argued that the centrale is absent in AMNH 9502, in contrast to Gow's [31] description for *Prolacerta*. However, according to Colbert [42], this bone had likely been lost during fossilization. *Prolacerta* specimen UWBM 95529 [32] preserves a centrale, in agreement with the initial statement by Gow [31], therefore, similar to what is observed in *Elessaurus gondwanoccidens*. No centrale bones appear to be present in *Tanystropheus longobardicus* (MCSN V 3730), although the distal tibial articular surface is wide. In *Tanystropheus*, the astragalar body and centrale thus are possibly indistinguishably fused [4]. The presence of a cartilaginous centrale in *Tanystropheus* also remains highly conjectural [13]. *Langobardisaurus* and *Macrocnemus* have the area distal and/or medial to the astragalus occupied by an ossified centrale [13]. According to Rieppel [15], in *Macrocnemus*, the tibia articulates with the astragalus, bearing a distinct articular facet on its medial side. This facet forms the proximal part of an embayment completed by the centrale and distal tarsal I, which accommodates the tibia during the stride phase when maximal propulsive force is applied. In *Dinocephalosaurus orientalis* the tarsus preserves three ossifications of generally rounded outlines. A small ossification is present between the astragalus and the calcaneum; it presumably corresponds to the fourth distal tarsal [33]. The proximal part of the ankle of *Boreopricea* (PIN 3708/1) consist of four elements, the centrale, the astragalus, the distal tarsal IV and the calcaneum. A foramen between the astragalus and the calcaneum is apparently missing in this specimen [36].

In *Elessaurus gondwanoccidens*, astragalus and calcaneum are unfused and lack a perforating foramen. This contrasts with *Prolacerta* [32], in which it is possible to observe a perforating foramen between these elements (UWBM 95529). A perforating foramen is absent in *Langobardisaurus* (MCSNB 2883, MCSNB 4870, MFSN 1921, MFSN 26829) and *Tanytrachelos* (VMNH 120015, YPM 8600), although this may be the consequence of small size, or even an artifact of preservation [5,45]. According to Rieppel *et al.* [46] this foramen is absent in *Tanystropheus cf. Ta. longobardicus*. A larger perforating foramen is present in *M. bassanii* and *Amotosaurus* [5,15,17]. The surface for tibial articulation of the calcaneum is slightly rounded,

and the articular surface of the distal tarsal IV is concave. The articular surface for the astragalus appears to be continuous with the articulation of the distal tarsals.

The calcaneum is quadrangular in lateral view, being wider on its anteroposterior axis than proximo-distally, becoming "L"-shaped distal to the fibula. The proximal surface of the calcaneum is marked by the presence of a rough tuberosity, the calcaneal tuber. In proximal view, the calcaneal tuber is "square"-shaped, longer proximo-distally than dorsoventrally, and the distal part presents a curvature. The tuber is proximo-distally longer than dorsoventrally tall, in similar proportions to that observed in most early archosauromorphs (e.g., *Tanytrachelos ahynis*, GR 306; *Trilophosaurus buettneri*, TMM 31025–140; *Azendohsaurus*, FMNH PR 2776) [4]. There is a notch between the main body of the calcaneum and the tuber. Among tanystropheids, the calcaneal tuber is only present in *Tanytrachelos ahynis* Olsen, 1979, although it is a typical characteristic of several clades within Archosauriformes. Benton and Allen [36] described a lateral tuber in *Boreopricea funerea* (PIN 3708/1). This element is almost rectangular, curves slightly upwards and both ventral and dorsal surfaces are smooth and slightly concave.

According to Nesbitt *et al.* [4], a fibular facet continuous with the lateral tuber is present in some *Trilophosaurus* (AMNH FARB 30836), *Proterosuchus alexanderi* Hoffman, 1965 (MCZ 4301), and *Erythrosuchus africanus* (NHMUK R3592). Pritchard *et al.* [5] assigned specimen GR 306 to *Tanytrachelos* based on features of the calcaneum, such as the strongly laterally expanded distal end and the distal curvature of the calcaneal tuber. The calcaneal tuber of GR 306 is still larger than those observed in other taxa. This structure is well evident in *Elessaurus gondwanoccidens* and compatible with that observed in GR 306, as the proximal surface of the calcaneum is marked by the development of its lateral margin, characterizing a rough tuberosity. Nesbitt *et al.* [4] noted a similar condition to *Tanytrachelos ahynis* (GR 306) in *Azendohsaurus* (FMNH PR 2776).

## Phylogenetic analyses

Our first analysis recovered two MPTs with 1104 steps (consistency index 0.325 and retention index 0.643), in which *Elessaurus gondwanoccidens* is the sister taxon of Tanystropheidae (Fig 5), the latter being a node-based clade comprising the most recent common ancestor of *Macrocnemus*, *Tanystropheus*, and *Langobardisaurus* and all its descendants [47]. *Elessaurus gondwanoccidens* shares some similar character states with Tanystropheidae, as the pleurapophysis of the second sacral vertebra distally bifurcated (character-state 131:0→1), tibia and fibula with a total length slightly larger than the femur (character-state 516:0→2, [2]) and "hook"-shaped fifth metatarsal (character-state 197:0→1). The node (*Elessaurus gondwanoccidens* + Tanystropheidae) is supported by the presence of a distally sharp posterior process of the second sacral rib, and the transverse process of the anterior caudal vertebrae angled posterolaterally. In addition, the new specimen presents some features only found in more specialized representatives within Tanystropheidae, such as the presence of a well-developed calcaneal tuber with a rough lateral margin.

Our second analysis, which included *Jesairosaurus lehmani* and *Dinocephalosaurus orientalis* as OTUs (see above), resulted in 13 equally parsimonious trees, each one with 1150 steps. In this second analysis, *Elessaurus* adopts different positions among the MPTs, it is recovered, e.g. within Archosauriformes, as a sister-taxa of Allokotosauria+Archosauriformes and an early rhynchosaur. The strict consensus of this alternative analysis depicts a large polytomy that includes *Elessaurus*, as well as most sampled archosauromorphs (S1 Fig). Although well-established clades, such as Rhynchosauria were not recovered by the strict consensus topology, the Tanystropheidae was consistently found as monophyletic. Most interestingly,

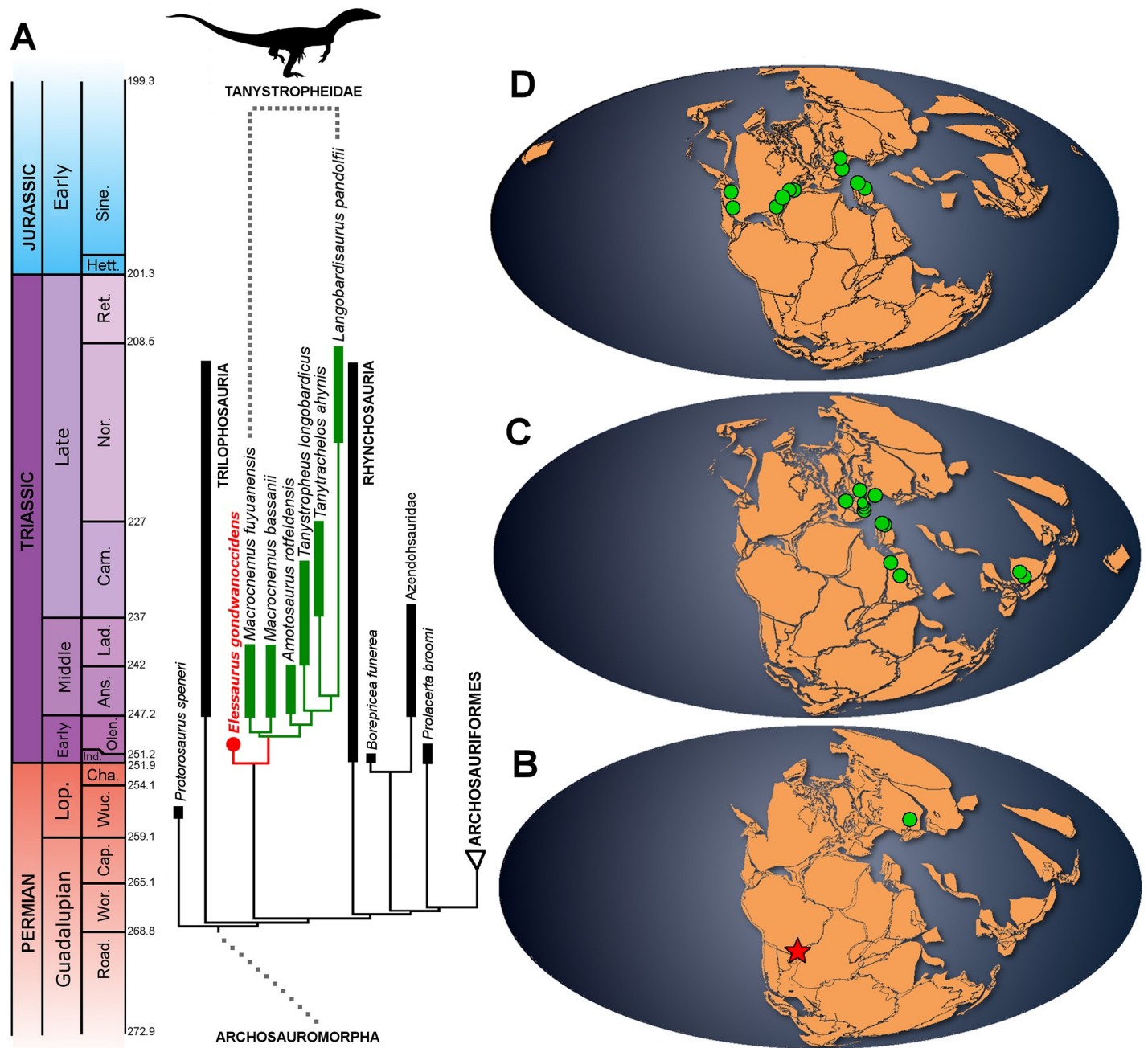

**Fig 5. A- Archosauromorph phylogeny showing the recovered position of *Elessaurus gondwanoccidens* (UFSM 11471), from the matrix of Pritchard *et al*.** [6] and the geographic distribution maps for Tanystropheidae through time (green circles) and the Brazilian fossil record (red star) (data from the Paleobiology Database, https://paleobiodb.org/#/) (B, Early Triassic; C, Middle Triassic; D, Late Triassic).

(*Jesairosaurus* + *Dinocephalosaurus*) has a sister-group relationship with the clade formed by the remaining archosauromorphs. Albeit this unusual position may reflect the protocol we employed to include both taxa as OTUs in Pritchard *et al*. [6] dataset (as well as the fact that we were not able to first-hand analyze relevant material), we should note that our second analysis may indicate that *Elessaurus* was more closely related to tanystropheids than to *Jesairosaurus* and *Dinocephalosaurus*.

## Discussion

### Taxonomic remarks

The specimen herein described is morphologically compatible with non-archosauriform arch-osauromorphs, and a close relationship with Tanystropheidae is supported by several characters (see above).

Recent phylogenetic analyses recovered tanystropheids and *Jesairosaurus lehmani* more closely related to each other than to other archosauromorphs[2]. Albeit the matrix of Pritchard *et al.* [6], used in this work, does not include *Jesairosaurus lehmani* (see above), *Elessaurus gondwanoccidens* differs from *J. lehmani* in several characters. According to Ezcurra [2], it is not possible to observe the presence of an internal trochanter or fourth trochanter in *J. lehmani*. Besides, the calcaneum of *J. lehmani*, although poorly preserved, lacks a calcaneal tuber (present in *Elessaurus gondwanoccidens*), and the distal end of the femur does not taper distally in dorsal view. The presence of the mentioned characters may indicate that *Elessaurus gondwanoccidens* is more closely related to Tanystropheidae than to *J. lehmani*. Among basal archo-sauromorphs, tibia/fibula longer than the femur is observed in Tanystropheidae and *Prolacerta*. Although *Prolacerta* is one of the best represented early archosauromorphs, its postcranial morphology remains poorly known, and most studies have focused on cranial anatomy. Its postcranial anatomy, however, was discussed by Gow [31], Colbert [40] and, more recently, Spiekman [32]. Despite the overall similarity among *Elessaurus gondwanocci-dens*, Tanystropheidae (e.g. *Macrocnemus*), and *Prolacerta*, the specimen herein described is more consistent with Tanystropheidae than with *Prolacerta*. A foramen perforating the astrag-alus/calcaneum is present in *Prolacerta* [32], whereas *Elessaurus gondwanoccidens* does not exhibit this feature. The good preservation of the proximal tarsals in *Elessaurus gondwanocci-dens* suggests that the absence of this foramen is not a preservation bias. Moreover, the poste-rior process of the bifurcated second sacral rib of *Elessaurus gondwanoccidens* is distally sharp, whereas in *Prolacerta* this process is blunt [6].

*Teyujagua paradoxa* Pinheiro *et al.* 2016 lacks comparable elements with *Elessaurus gond-wanoccidens*, this species was recovered as the sister taxon to the Archosauriforms and is thus more closely related to proterosuchids than *Prolacerta* and, consequently, *Elessaurus gondwa-noccidens* [23].

### Biogeographical implications

In the aftermath of the Permian–Triassic crisis, earliest Triassic continental communities were extremely impoverished, including few small and unspecialized tetrapod taxa [8,24,48]. The adaptive radiation of early archosauromorphs, including tanystropheids, possibly occurred already during the Early Triassic. Although the most abundant and the better-known records of this group belong to the Middle Triassic (Ladinian) of Switzerland and Italy, this group also has rare records in Lower Triassic strata. *Amotosaurus rotfeldensis* and *Augustaburiania vata-gini* are thus far the earliest nominal taxa of Tanystropheidae, being known from non-marine rocks from Germany and Russia [5,8,17].

Among the genera abovementioned, only *Macrocnemus* and *Tanystropheus* are known to occur in both the western and eastern Tethyan province, with specimens of *M. aff. fuyuanensis* and *T. longobardicus* from Europe being slightly older (late Anisian to early Ladinian) than Chinese ones (Ladinian) [14]. Jaquier *et al.* [14] and Pritchard *et al.* [5] proposed that during the late Early Triassic, tanystropheid reptiles first evolved from their late Permian and Early Triassic ancestors in central Pangea and dispersed afterwards along the western and eastern margins of the Tethys Ocean during the Middle Triassic.

*Elessaurus gondwanoccidens*, together with the specimens reported by De-Oliveira *et al.* [18], comprises the known record of Tanystropheidae-like archosauromorphs in South America. Albeit the tanystropheid remains reported by De-Oliveira *et al.* [18] may correspond to *E. gondwanoccidens*, a direct comparison is hindered by the fact that previous to the discovery of this latter, only cervical vertebrae were recovered. The recovery of *Elessaurus gondwanoccidens* close to Tanystropheidae, suggests the diversification of tanystropheid-related animals in South America still during the Early Triassic. Furthermore, it corroborates an early diversification of the group in central Pangea, possibly with a Gondwanan origin, reaching cosmopolitan distribution already during the early Mesozoic.

The transition of tanystropheids from terrestrial to semiaquatic and, then, aquatic and even marine habitats throughout the Triassic was probably connected with the diversification of the terrestrial biota, niche packing, increasing competitive pressure within continental communities, and diversification of predators [8]. *Elessaurus gondwanoccidens* may represent one of the oldest Tanystropheidae-related archosauromorphs to this date. The fact that the new specimen was collected in a depositional environment of ephemeral fluvial systems in an arid landscape, quite distinct from the marine deposits where tanystropheids are usually found, further corroborates the ecological plasticity of the clade.

## Tanystropheid ecology and the lifestyle of *Elessaurus gondwanoccidens*

Tanystropheidae is a clade mainly characterized by a long neck formed by elongate cervical vertebrae with low neural spines [5,8,13,39]. Some tanystropheids possibly inhabited terrestrial or semi-aquatic habitats, whereas more specialized forms may have been fully aquatic. Tanystropheid lifestyle, however, is still a matter of controversy. *Macrocnemus* and *Langobardisaurus* were supposedly terrestrial [9,15,49]. The association of *Tanytrachelos ahynis* with numerous fossil fishes, a lacustrine insect assemblage, abundant branchiopods, and phyllocarids, suggests it was aquatic, living in freshwater environments [34]. Some researchers indicate a digitigrade stance in the pes of *Tanytrachelos* [50], *Macrocnemus* [51] and *Langobardisaurus* [49], and a possibly bipedal posture, during rapid locomotion–as previously stated by Rieppel [15] for *Macrocnemus*–or even while standing and walking [12]. The foot of *Macrocnemus* appears to suit terrestrial locomotion, a conclusion further supported by the structure of its pelvic girdle [15]. The lifestyle of *Tanystropheus*, the largest and most bizarre of all tanystropheids, is still debatable. Recent osteological analyses do not support a fully aquatic habit for this animal [52], even though many skeletal features indicative of terrestrial habits in *Macrocnemus* are absent in *Tanystropheus*. For instance, it does not present bifurcating pleurapophyses on the second sacral vertebra; the preacetabular process is absent in the ilium; tarsal ossifications show a greater degree of reduction; the hooked fifth metatarsal is less distinctly differentiated and, finally, the metatarsus is far less asymmetrical [15]. One of the most striking features would be the much longer neck present in *Tanystropheus*. Renesto [12] emphasized that the neck of *Tanystropheus* was rather mobile and held horizontally or considerably raised. On the other hand, Tschanz [11], drawing a comparison with extant reptiles (*Iguana* and *Varanus*), concluded that the neck of *Tanystropheus* would be almost inflexible, indicating a fully aquatic habit. Moreover, the reduced size of the forelimb suggests that it did not have a major contribution to any kind of locomotion [12,13]. Reassessing this genus, Nosotti [13] regarded *Tanystropheus* as an aquatic animal with close terrestrial ancestors, living in shallow waters and probably returning to land for reproduction. Recently, some authors have proposed that the locomotion of *Tanystropheus* is consistent both with feeding on aquatic prey and with a semi-aquatic lifestyle in the near-shore environments [9].

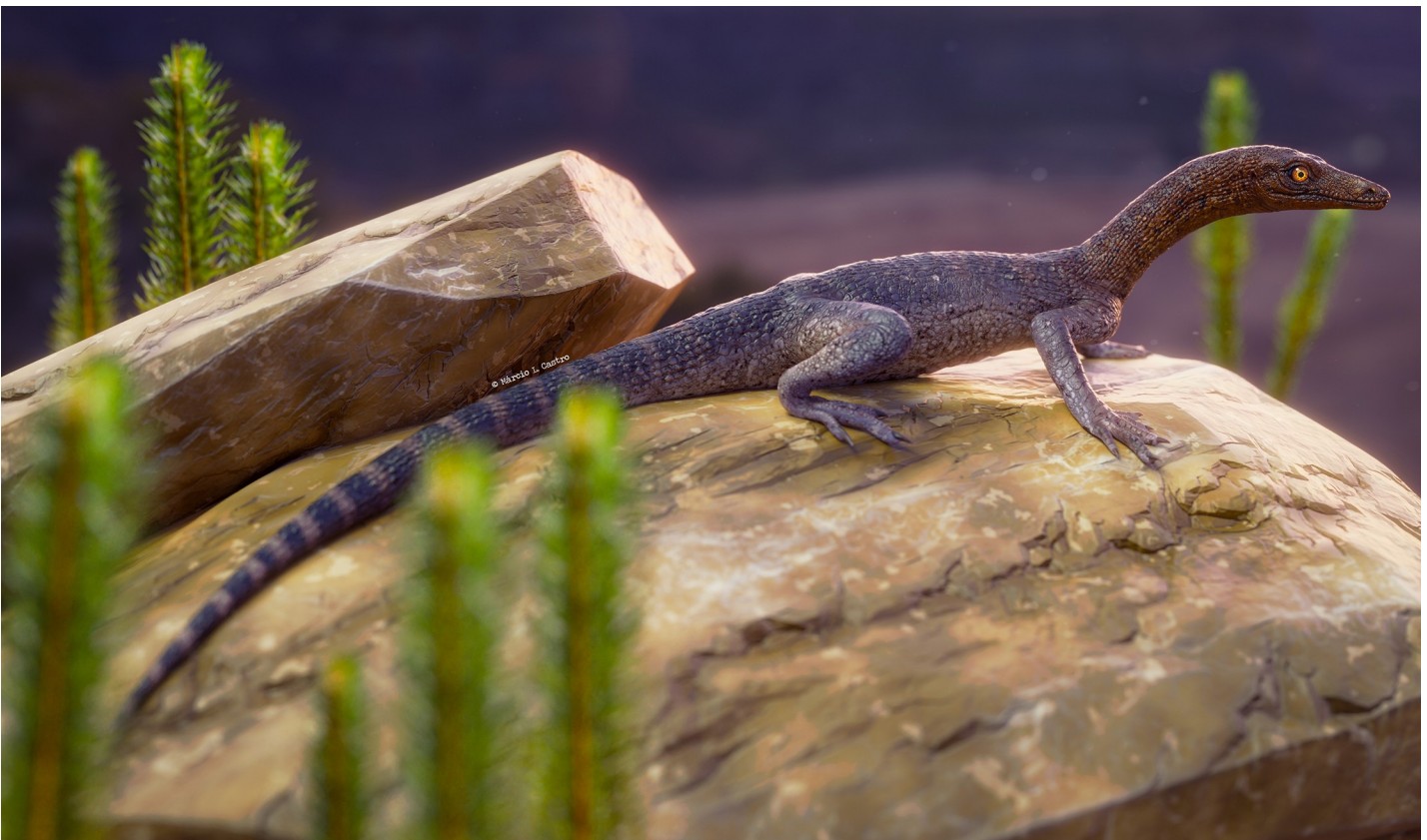

**Fig 6. Life restoration of *Elessaurus gondwanoccidens*, from the Sanga do Cabral Formation (Lower Triassic), Brazil.** Artwork by Márcio L. Castro.

Liu *et al.* [7] described a new specimen of the aquatic reptile *Dinocephalosaurus* (LPV 30280) from the Middle Triassic of South China containing an embryo in the abdominal region. The features observed in this specimen and the limb skeleton of *Dinocephalosaurus*, as well as the proportions of the limbs, indicates that amongst all archosauromorphs related to tanystropheids it is the taxon most highly adapted to a marine habitat, although functional considerations indicate that *Tanystropheus* was most probably marine as well [7,33].

The skeletal anatomy of *Tanystropheus* is unique, and there are no analogs in present-day or extinct animals. Its peculiar body plan, together with its impressive overall size (the largest individuals of *T. longobardicus* reached up to five meters in length) renders this animal a weird appearance, being still a palaeoecological and functional enigma [9].

At least one specimen of *Tanytrachelos* (AMNH FARB 7206) [5] bears a calcaneal tuber similar to that one of *Elessaurus gondwanoccidens*. Some authors have argued for the absence of calcaneal tuber in tanystropheids (e.g., [13,15]). The absence of this tuber in more specialized forms might be attributable to an aquatic lifestyle [5,15]. Based on the presence of a calcaneal tuber, the hooked fifth metatarsal and the distally bifurcating pleurapophyses on the second sacral vertebra, we propose a terrestrial habit for *Elessaurus gondwanoccidens* (Fig 6), similar to what is argued for *Macrocnemus* and *Tanytrachelos*, and distinct from what is usually proposed for *Tanystropheus*. This interpretation agrees with some authors [5,8,13] in which tanystropheids or close relatives were able to inhabit a wide range of climatic conditions and that, although possessing most of its representatives with affinities to an aquatic lifestyle, this group possibly presents close ancestors with terrestrial habit.

The depositional model of the locality from where *Elessaurus gondwanoccidens* was collected also supports the interpretation based on its morphology. The Sanga do Cabral Formation is characterized as a system of ephemeral, high energy river channels with wide and extensive alluvial plains, containing a rich assemblage of terrestrial and aquatic tetrapods composed of temnospondyls, procolophonoids, and archosauromorphs [21,22,53]. Considering both morphology and the environment described for the locality of Bica São Tomé, this animal would probably be terrestrial, inhabiting the vicinities of shallow waters and low-sinuosity river environments.

## Conclusions

Until recently, the Sanga do Cabral Formation provided only few remains assigned to Archosauromorpha indet. Now, at least two independent lineages were reported for this unit ([18,22,23,54] and this work). Although rare, these fossils demonstrate that archosauromorphs had already diversified in the Early Triassic of western Gondwana. *Elessaurus gondwanoccidens* is here recovered as the sister taxon of Tanystropheidae and was collected from rocks reminiscent of continental environments dominated by ephemerous water bodies. Most representatives of Tanystropheidae (e.g. *Tanystropheus*) belong to marine environments. The results of the present work suggest that a terrestrial mode of life was plesiomorphic for Tanystropheidae and maintained by some of its representatives (e.g. *Macrocnemus*). The record of Tanystropheidae-related taxa in Permian and Lower Triassic layers from South America indicates a premature wide distribution of this clade, with a possible Gondwanan origin.

## Supporting information

**S1 Fig. Strict consensus of the phylogenetic analysis including *Jesairosaurus lehmani* and *Dinocephalosaurus orientalis*, in the matrix of Pritchard et al. 2018.**
(DOCX)

**S1 Material.**
(NEX)

## Acknowledgments

For allowing access to fossil collections, FLP is indebted to Christian Klug (Paläontologisches Institut und Museum, Universität Zürich), Rainer Schoch (Naturkunde Museum Stuttgart), Oliver Rauhut and Markus Moser (Bayerische Staatssammlung für Paläontologie und Geologie), Mark Norell and Carl Mehling (American Museum of Natural History), Sandra Chapman and Lorna Steel (Natural History Museum), Bernhard Zipfel and Bruce Rubidge (Evolutionary Studies Institute, University of Witwatersrand), Claire Browning and Zaituna Skosan (Iziko South African Museum). Rodrigo Müller and Flavio Pretto (Centro de Apoio à Pesquisa Paleontológica da Quarta Colônia, Rio Grande do Sul, Brazil) made useful comments on description of the specimen and Lívia Roese Miron authored the anatomical drawings. We also thank Reinaldo José Lopes for etymologic assistance, Martin Ezcurra and an anonymous referee for valuable contributions that improved the original manuscript.

## Author Contributions

**Methodology:** Tiane M. De-Oliveira, Felipe L. Pinheiro, Átila Augusto Stock Da-Rosa, Sérgio Dias-Da-Silva, Leonardo Kerber.

**Supervision:** Felipe L. Pinheiro, Leonardo Kerber.

**Writing – original draft:** Tiane M. De-Oliveira.

**Writing – review & editing:** Felipe L. Pinheiro, Átila Augusto Stock Da-Rosa, Sérgio Dias-Da-Silva, Leonardo Kerber.

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
