## [Decision Letter · Decision Letter 0]

6 Aug 2019

PONE-D-19-19097

A NEW ARCHOSAUROMORPH FROM SOUTH AMERICA PROVIDES INSIGHTS ON THE EARLY DIVERSIFICATION OF TANYSTROPHEIDS

PLOS ONE

Dear De-Oliveira,

Thank you for submitting your manuscript to PLOS ONE. After careful consideration, we feel that it has merit but does not fully meet PLOS ONE’s publication criteria as it currently stands. Therefore, we invite you to submit a revised version of the manuscript that addresses the points raised during the review process.

We would appreciate receiving your revised manuscript by Sep 20 2019 11:59PM. To enhance the reproducibility of your results, we recommend that if applicable you deposit your laboratory protocols in protocols.io, where a protocol can be assigned its own identifier (DOI) such that it can be cited independently in the future. For instructions see: http://journals.plos.org/plosone/s/submission-guidelines#loc-laboratory-protocols

We look forward to receiving your revised manuscript.

Kind regards,

Jörg Fröbisch, Ph.D.

Academic Editor

PLOS ONE

2. In your manuscript, please provide additional information regarding the specimens used in your study. Ensure that you have reported specimen numbers and complete repository information, including museum name and geographic location.

For more information on PLOS ONE's requirements for paleontology and archaeology research, see " ext-link-type="uri" xlink:type="simple">https://journals.plos.org/plosone/s/submission-guidelines#loc-paleontology-and-archaeology-research."

3. Please take this opportunity to be sure you have met all of our guidelines for new species. For proper registration of a new zoological taxon, we require two specific statements to be included in your manuscript.

a)    In the Results section, the globally unique identifier (GUID), currently in the form of a Life Science Identifier (LSID), should be listed under the new species name, for example:

Anochetus boltoni Fisher sp. nov. urn:lsid:zoobank.org:act:B6C072CF-1CA6-40C7-8396-534E91EF7FBB

Another LSID for the manuscript itself should also appear within the Nomenclature statement. You will need to contact Zoobank (zoobank.org/About) to obtain a GUID (LSID). You should receive one LSID for your manuscript and a separate, unique LSID for the new species.

b)    Please also insert the following text into the Methods section, in a sub-section to be called "Nomenclatural Acts":

The electronic edition of this article conforms to the requirements of the amended International Code of Zoological Nomenclature, and hence the new names contained herein are available under that Code from the electronic edition of this article. This published work and the nomenclatural acts it contains have been registered in ZooBank, the online registration system for the ICZN. The ZooBank LSIDs (Life Science Identifiers) can be resolved and the associated information viewed through any standard web browser by appending the LSID to the prefix "" ext-link-type="uri" xlink:type="simple">http://zoobank.org/". The LSID for this publication is: urn:lsid:zoobank.org:pub: XXXXXXX. The electronic edition of this work was published in a journal with an ISSN, and has been archived and is available from the following digital repositories: PubMed Central, LOCKSS [author to insert any additional repositories].

All PLOS ONE articles are deposited in PubMed Central and LOCKSS. If your institute, or those of your co-authors, has its own repository, we recommend that you also deposit the published online article there and include the name in your article.

Following a recent ruling by the International Commission on Zoological Nomenclature, electronic journals are now a valid format for publication of new zoological taxa. In order to ensure the valid publication of your new species, please be sure to include the updated version of Nomenclatural Acts (above). A complete explanation of our guidelines for publishing new species can be found on our website: http://www.plosone.org/static/guidelines#zoological.

a) You may seek permission from the original copyright holder of Figure 1 to publish the content specifically under the CC BY 4.0 license.  

Additional Editor Comments (if provided):

Please pay attention to the comments and requests of both reviewers, in particular with regards to the phylogenetic analysis and inclusion of crucial additional (and previously included) basal tanystropheids and close relatives.

Reviewers' comments:

Reviewer's Responses to Questions

**Comments to the Author**

1. Is the manuscript technically sound, and do the data support the conclusions?

Reviewer #1: No

Reviewer #2: Yes

2. Has the statistical analysis been performed appropriately and rigorously? 

Reviewer #1: N/A

Reviewer #2: N/A

3. Have the authors made all data underlying the findings in their manuscript fully available?

Reviewer #1: Yes

Reviewer #2: Yes

4. Is the manuscript presented in an intelligible fashion and written in standard English?

Reviewer #1: Yes

Reviewer #2: Yes

5. Review Comments to the Author

Reviewer #1: The manuscript reported a new species related with tanystropheid protorosaurs from South America and is an important addition to the scientific world. The manuscript is also generally well written and well organized. However, complete omission of two important protorosaurs including Boreopricea and Dinocephalosaurus throughout the manuscript prevents me to recommend the publication of the manuscript in its present form. One of the major conclusions of the manuscript is that the new species is the sister taxon of tanystropheid protorosaurs. However, two other well known protorosaur taxa including Jesairosaurus and Dinocephalosaurus were not included into the phylogenetic analysis, while they have been recovered as the sister group of tanystropheids in previous publications. See below for detailed comments.

1. p. 3, l. 47-50: Any reason to exclude Dinocephalosaurus from Tanystropheidae? Dinocephalosaurus has been recovered as a tanystropheid by Liu et al. (2017, Nature Communications) and its morphology has been also well established by Liu et al. (2017) and Rieppel et al. (2008, JVP).

2. p. 5, Phylogenetic Analysis: this section is problematic. The authors exclude Jesairosaurus from phylogenetic analysis simply because they did not examine the specimens by themselves. However, I noticed that the same group of authors indeed included Jesairosaurus into the phylogenetic analysis (De Oliveira et al, 2018, APP). Especially De Oliveira et al. (2018) noticed that Jesairosaurus is the immediate sister group of tanystropheids. Now the same group of authors described another protorosaur claiming a sister group relationship of the newly reported taxon and tanystropheids. This is apparently inconvincible. Indeed the morphology of Jesairosaurus was well presented by Jalil (1997, JVP). Anyway the authors are not preparing a new data matrix and they simply used an existed data matrix. I see no reason to exclude Jesairosaurus from their phylogenetic analysis.

Also Boreopricea and Dinocephalosaurus were recovered as the consecutive sister groups of the traditionally recognized tanystropheids (Liu et al., 2017). The authors even did not mention these two taxa throughout the manuscript, while morphology of both taxa has been well established (Benton and Allen, 1997, Palaeontology; Rieppel et al., 2008; Liu et al., 2017).

3. p. 7, diagnosis: any autapomorphy to support the species status of the new taxon?

4. Morphological comparison with archosaurmorphs: the authors should also compare their new taxon to the protorosaur Boreopricea and Dinocephalosaurus.

5. p. 18, l. 25: change “analyzes” to “analyses”.

6. p. 18-19, taxonomic remarks: still the reason given by the authors to exclude Jesairosaurus from their phylogenetic analysis is not convincible. Also the authors should give some remarks about their new taxon and those reported by the same authors (De Oliveira et al, 2018).

7. p. 21, Tanystropheid ecology and the lifestyle of Elessaurus gondwanoccidens: the authors apparently should mention Dinocephalosaurus since it’s ecology and lifestyle has been studied in very details.

Reviewer #2: The authors describe a new genus and species of early archosauromorph from the Early Triassic of Brazil. The specimen is extremely important for the fossil record of the group in South America because of the scarcity of Early Triassic specimens in this continent. The description of the new species is very detailed and the anatomy is well figured. The phylogenetic analysis has been well conducted, but I have two comments that I would like that the authors address in the revised version of the manuscript. As a result, I strongly recommend the acceptance of this manuscript after minor changes indicated in a edited PDF version of the manuscript and the two moderate comments that I detailed as follows:

Main comments:

- Page 11, Phylogenetic analysis: The authors use a modified version of the data matrix published by Pritchard et al. (2015) to test the phylogenetic relationships of the new taxon. The authors justify properly the selection of this data set claiming that it is the analysis that includes a broader sample of tanystropheids. Nevertheless, the taxon and character sample of this data matrix considerably differs from that of Ezcurra (2016) and it would be very useful to test if the position of the new taxon as a sister taxon of Tanystropheide is also recovered in the latter data set. The data matrix of Ezcurra (2016) includes Jesairosaurus (a probable sister taxon to Tanystropheidae) and several phylogenetically informative characters among non-archosaurian archosauromorphs that are absent in the analysis of Pritchard et al. (2015). As a result, I strongly suggest the authors to conduct both analyses, using the latest versions of the data matrices of Pritchard et al. (2015) and Ezcurra (2016), respectively.

- Page 11, lines 116–117: I can't see the utility of this additional analysis. If you are doing a resample analysis it makes sense to check the percentage of recovery of the nodes because they are based on a perturbation and reanalysis of the data matrix. But if you are changing the number of steps that have the retained trees would not be informative the percentages of recovery. It is an analogous case to that of using a majority rule consensus tree from most parsimonious trees (you are wrongly excluding equally parsimonious topologies).

Yours sincerely,

Martin Ezcurra

6. PLOS authors have the option to publish the peer review history of their article (what does this mean?). If published, this will include your full peer review and any attached files.

Reviewer #1: No

Reviewer #2: Yes: Martín D. Ezcurra

---

## [Author Response · Author response to Decision Letter 0]

30 Oct 2019

Dear Dr. Jörg Fröbisch

Academic Editor

PLOS ONE

We provide below a detailed point-by-point answer to reviewer’s comments of our manuscript PONE-D-19-19097 ‘A new archosauromorph from South America provides insights on the early diversification of tanystropheids’. 

Reviewer #1

We thank the reviewer for its thorough review of the text, which substantially improved our MS presentation and content. In this respect, the reviewer’s corrections were fully adopted in the new version.

Our responses to the main concerns of Reviewer #1 are provided below:

- The revised version includes comparisons with both Boreopricea and Dinocephalosaurus;

- We did not exclude any taxon from the original dataset of Pritchard et al. (2015). In fact, the analysis of De-Oliveira et al. (2018) (mentioned by Reviewer #1) used a completely different data matrix – an updated version of Ezcurra (2016) dataset.

- Jesairosaurus and Dinocephalosaurus were included as OTUs in the dataset of Pritchard et al. (2015) and the results of this analysis is discussed and illustrated in the revised MS and supplementary material. 

- The new taxon is the diagnosed by a combination of characters. The only potential autapomorphy (a lateral groove on the distal surface of the tibia) is probably taphonomic in origin, and this is discussed in the text. 

- Some remarks were made about the potential correlation between Elessaurus and the tanystropheid vertebrae reported by De-Oliveira et al. (2018).

- The ecology of Dinocephalosaurus is briefly discussed. 

Reviewer #2

The main issues raised by Dr. Martin Ezcurra concern the phylogenetic analysis methodology:

- We agree with Dr. Ezcurra in that our second analysis is superfluous and completely excluded it from the manuscript. 

- Albeit we exploratively included Elessaurus in the dataset of Ezcurra (2016), we choose not to depict it in the revised version for the reasons below:

1. The original dataset of Ezcurra (2016) has a low tanystropheid representativity, as this study was designed to test the relationships among main archosauromorph clades, with emphasis on early archosauriforms. An updated version of the dataset was provided by Ezcurra Butler (2018), which included score propositions for a number of tanystropheids that were absent from the original work. However, the a posteriori addition of new OTUs resulted in instability of the dataset as a whole, as the data matrix of Ezcurra Butler (2018) was designed for morphological disparity analyses rather than for testing phylogenetic relationships. As such, we consider the data matrix of Pritchard et al. (2015) more suitable for our purposes. 

2. Inclusion of Elessaurus in the dataset of Ezcurra (2016) recovered an apparently aberrant topology, in which the new taxon is recovered in a position we judge incompatible with its morphology. Tracking the character states that would sustain a derived position for Elessaurus in this explorative analysis, we noted the decisive influence of the presence and morphology of a calcaneal tuber and the hook-shaped metatarsal V. Although in the dataset of Ezcurra (2016) these characters are recovered as synapomorphic for Crocopoda, non-sampled unambiguous tanystropheids can also bear these features in a derived condition (e.g. Tanytrachelos, see Pritchard et al., 2015). As we thoroughly discuss in our manuscript, although the new specimen being incomplete, the morphology of Elessaurus is only fully compatible with tanystropheid-like archosauromorphs, and this relationship is in agreement with the dataset of Pritchard et al. (2015), this later including a broader sample of tanystropheids (and also Tanytrachelos). We, thus, prefer to stick with Pritchard’s dataset in combination with our comparative anatomical analyses. 

We thank for the time dispensed in editing and reviewing our work.

Yours sincerely

Tiane De-Oliveira

Universidade Federal do Pampa

Universidade Federal de Santa Maria

---

## [Decision Letter · Decision Letter 1]

23 Dec 2019

PONE-D-19-19097R1

A NEW ARCHOSAUROMORPH FROM SOUTH AMERICA PROVIDES INSIGHTS ON THE EARLY DIVERSIFICATION OF TANYSTROPHEIDS

PLOS ONE

Dear De-Oliveira,

Thank you for submitting your manuscript to PLOS ONE. After careful consideration, we feel that it has merit but does not fully meet PLOS ONE’s publication criteria as it currently stands. Therefore, we invite you to submit a revised version of the manuscript that addresses the points raised during the review process.

We would appreciate receiving your revised manuscript by Feb 06 2020 11:59PM. To enhance the reproducibility of your results, we recommend that if applicable you deposit your laboratory protocols in protocols.io, where a protocol can be assigned its own identifier (DOI) such that it can be cited independently in the future. For instructions see: http://journals.plos.org/plosone/s/submission-guidelines#loc-laboratory-protocols

We look forward to receiving your revised manuscript.

Kind regards,

Jörg Fröbisch, Ph.D.

Academic Editor

PLOS ONE

Additional Editor Comments (if provided):

Dear authors, I acknowledge that you made all the changes, however it beomomes apparent from the second round of reviews that the reviewers (and I fully agree with them) would like to have a more expanded section treating the phylogenetic analyses and placement of the new taxon. Hence, I strongly recommend for you to include the Ezcurra (2016) analysis, as requested by Martin Escurra and potentially also test the position of the taxon in Liu et al. (2017). I'm looking forward to your revised version. Best wishes and happy holidays, Jörg Fröbisch

Reviewers' comments:

Reviewer's Responses to Questions

**Comments to the Author**

1. If the authors have adequately addressed your comments raised in a previous round of review and you feel that this manuscript is now acceptable for publication, you may indicate that here to bypass the “Comments to the Author” section, enter your conflict of interest statement in the “Confidential to Editor” section, and submit your "Accept" recommendation.

Reviewer #1: All comments have been addressed

Reviewer #2: (No Response)

2. Is the manuscript technically sound, and do the data support the conclusions?

Reviewer #1: Partly

Reviewer #2: Yes

3. Has the statistical analysis been performed appropriately and rigorously? 

Reviewer #1: N/A

Reviewer #2: N/A

4. Have the authors made all data underlying the findings in their manuscript fully available?

Reviewer #1: Yes

Reviewer #2: Yes

5. Is the manuscript presented in an intelligible fashion and written in standard English?

Reviewer #1: Yes

Reviewer #2: Yes

6. Review Comments to the Author

Reviewer #1: I noticed that the authors made substantial revisions taking into the two reviewers’ comments. However, I am still troubled by the phylogenetic analysis performed by the authors and feel that the phylogenetic evidence given by the authors are not convincible to support the sister relationship between their new taxon and Tanystropheidae, as they concluded. Ideally the authors should combine the two independent data matrix by Pritchard et al. (2015) and Liu et al. (2017) into one to test the phylogenetic position of the new taxon. Both are highly related with phylogenetic relationships of tanystropheids. If this is beyond the scope of the paper, it should be quite easy to test the position of the new taxon using the Liu et al. (2017) data matrix and compare the result with the one using Pritchard et al. (2015) data matrix. Also the supplementary figure should be included in the main text as another figure. I do not see the reason why this figure was put into the supplement. The authors should also clarify which data matrix exactly they used. In the author response letter, they indicated that they used the data matrix by Pritchard et al. (2015, JVP). In the revised manuscript, however, they cited the data matrix from Pritchard et al. (2018, Current Biology) for phylogenetic analysis. See below for other minor revisions.

1. l. 51: change “recovered more closer phylogenetically of” to “recovered closer phylogenetically to”.

2. l. 61: change “comes” to “come”.

3. Throughout the manuscript, I think that the authors should include another quite complete specimen of Dinocephalosaurus described by Liu et al. (2017) for comparison. There is no any reason to compare only with one specimen of a taxon when there is other good material of the same taxon available.

4. l. 301: what is Eucrocopoda??? An introduction should be given here, or at least a reference introducing this taxon. PLOS One is a multidiscplinary journal.

5. l. 444: change “Benton and Allen [36] described a lateral tuber in Boreopricea funerea (PIN 3708/1), this” to “Benton and Allen [36] described a lateral tuber in Boreopricea funerea (PIN 3708/1). This”.

6. l. 498: change “Recent phylogenetic analyzes recovered tanystropheids” to “Recent phylogenetic analyses recovered tanystropheids”.

7. l. 547: “The recovery of Elessaurus gondwanoccidens as the sister taxon of Tanystropheidae”. Apparently the authors do not provide convincible evidence to support this statement.

8. Supplementary Material 2 should be cited somewhere in the main text.

Reviewer #2: This is the revised version of a manuscript that I reviewed a few months ago. I found that the authors followed the vast majority of changes that I suggested, with the exception of including the new taxon in the most recent version of the phylogenetic dataset of Ezcurra (2016). Although I consider that the discussion of the phylogenetic relationships of the new taxon would benefit from its inclusion in the latter dataset, I consider that it is not completely necessary for the publication of the manuscript and it is something that others can explore in the future.

When I downloaded the data matrix provided as supplementary information I found that all characters were additive and scaled by 0.322 (I don’t know if this is the case, but this is an artefact that occurs when the data matrix is edited with Winclada and exported to other program). When I analysed the data matrix with this ordering of characters I recovered the same results reported by the authors (1 most parsimonious tree of 1120 steps). However, when I turned as additive only the characters considered as additive by Pritchard et al. (2018) I found 2 MPTs of 1104 (a considerably lower number of steps). However, only one difference in topology is present between the strict consensus trees of the first and second analyses.

Similarly, the “alternative” analysis of the authors recovered 30 MPTs of 1168 steps, but when I set the same ordered characters used by Pritchard et al. (2018) I recovered 13 MPTs of 1151 steps. Nevertheless, the strict consensus tree of this analysis is the same as that reported by the authors. The changes in the results of the phylogenetic analyses don’t alter the main results and conclusions of the manuscript, but it is something that the authors have to correct in the new version of the manuscript.

In addition, it would be interesting if the authors discuss the alternative positions that the new taxon adopt among the MPTs of the “alternative analysis” because it is found as the sister-taxon to Tanystropheidae but also as e.g. an early rhynchosaur, sister-taxon to Allokotosauria+Archosauriformes, and within Archosauriformes.

In conclusion, I think that the manuscript can be accepted after minor changes.

Yours sincerely,

Martin Ezcurra

7. PLOS authors have the option to publish the peer review history of their article (what does this mean?). If published, this will include your full peer review and any attached files.

Reviewer #1: No

Reviewer #2: Yes: Martin Ezcurra

---

## [Author Response · Author response to Decision Letter 1]

4 Feb 2020

Dear Dr. Jörg Fröbisch

Academic Editor

PLOS ONE

We provide below a detailed point-by-point answer to reviewer’s comments of our manuscript PONE-D-19-19097 ‘A new archosauromorph from South America provides insights on the early diversification of tanystropheids’. 

Reviewer #1

We thank the reviewer for its thorough review of the text, which were included in the new version. The suggestions substantially improved our manuscript presentation and content. 

Our responses to the main concerns of Reviewer #1 are provided below:

- According to the suggestion of Reviewer #1, we prefer to make our phylogenetic position of E. gondwanoccidens less emphatic, as an enigmatic archosauromorph closely related to tanystropheids, which is in accordance to morphology and the recovered phylogenetic position in the dataset of Pritchard et al. (2018). At the end of the current work, Elessaurus is treated as a sister-taxon of Tanystropheidae. This result is in accordance to the occurrence of fossils related to tanystropheids in Sanga do Cabral Formation (De-Oliveira et al. 2018).

- Elessaurus gondwanoccidens was thoroughly codified in the matrixes of Ezcurra (2016), Ezcurra Butler (2018) and Pritchard et al. (2018). We also included an additional analysis with the a posteriori inclusion of Dinocephalosaurus orientalis and Jesairosaurus lehmani, also in accordance with the reviewer’s requests. 

- The dataset of Liu et al (2017) has the same tanystropheid representativity to what is displayed by our second dataset (with the addition of Dinocephalosaurus and Jesairosaurus to the matrix of Pritchard et al. 2018). As such, we believe that the effort of scoring Elessaurus to Liu et al (2017) dataset wouldn’t add to the resolution of Elessaurus relationships. We should also note that the dataset of Liu et al. (2017) is a composite of three previous data matrixes: Benton et al. (1997), Jalil (1997) and Dilkes (1998). Despite these being seminal works dealing with archosauromorph phylogeny, much has been done after these three contributions, and the datasets of Pritchard et al. (2018) and Ezcurra (2016) are much more reliable and up-to-date than previous works.

- As mentioned earlier, the original dataset of Ezcurra (2016) has a low tanystropheid representativity. The updated version of the dataset provided by Ezcurra Butler (2018), resulted in instability of the dataset as a whole, as the data matrix of Ezcurra Butler (2018) was designed for morphological disparity analyses rather than for testing phylogenetic relationships. As such, we consider the data matrix of Pritchard et al. (2018) more suitable for our purposes.

- As we thoroughly discuss in our manuscript, although the new specimen is incomplete, the morphology of Elessaurus is only fully compatible with tanystropheid-like archosauromorphs, and this relationship is in agreement with the dataset of Pritchard et al. (2018), this later including a broader sample of tanystropheids (and also Tanytrachelos). Again, we should highlight that the tanystropheid affinities of Elessaurus is in agreement with the Sanga do Cabral Fm fauna (De-Oliveira et al. 2018). 

Reviewer #2

The main issues raised by Dr. Martin Ezcurra concern the phylogenetic analysis methodology:

- The changes in the results of phylogenetic analyses have been added to the text, as well, the discussion of the alternative positions that the new taxon adopt among the MPTs of the second analysis.

- We agree that other phylogenetic analyses are not completely necessary for the publication of the manuscript, and it is something that could be explored in future contributions. 

We thank for the time dispensed in editing and reviewing our work.

Yours sincerely

Tiane De-Oliveira

Universidade Federal do Pampa

Universidade Federal de Santa Maria

---

## [Editor Report · Decision Letter 2]

3 Mar 2020

PONE-D-19-19097R2

A NEW ARCHOSAUROMORPH FROM SOUTH AMERICA PROVIDES INSIGHTS ON THE EARLY DIVERSIFICATION OF TANYSTROPHEIDS

PLOS ONE

Dear De-Oliveira,

Thank you for submitting your manuscript to PLOS ONE. After careful consideration, we feel that it has merit but does not fully meet PLOS ONE’s publication criteria as it currently stands. Therefore, we invite you to submit a revised version of the manuscript that addresses the points raised during the review process.

We would appreciate receiving your revised manuscript by Apr 17 2020 11:59PM. To enhance the reproducibility of your results, we recommend that if applicable you deposit your laboratory protocols in protocols.io, where a protocol can be assigned its own identifier (DOI) such that it can be cited independently in the future. For instructions see: http://journals.plos.org/plosone/s/submission-guidelines#loc-laboratory-protocols

We look forward to receiving your revised manuscript.

Kind regards,

Jörg Fröbisch, Ph.D.

Academic Editor

PLOS ONE

Additional Editor Comments (if provided):

Dear authors, this manuscript is essentially about to be accepted. I made some very minor (mostly wording) changes to the manuscript text file and would like you to review these and preferable accept them. Please subsequently upload the clean modified file to the system. Also I noted that part of the stratigraphic chart of Fig. 5 is not in English (equivalent of Early, Middle, Late), please replace the Spanish abbreviations with English ones und upload the corrected version. Afterwards I will be happy to accept this manuscript for publication. Thank you very much and best wishes

---

## [Author Response · Author response to Decision Letter 2]

5 Mar 2020

All changes were made in accordance to the editor's recommendations.

---

## [Editor Report · Decision Letter 3]

12 Mar 2020

A NEW ARCHOSAUROMORPH FROM SOUTH AMERICA PROVIDES INSIGHTS ON THE EARLY DIVERSIFICATION OF TANYSTROPHEIDS

PONE-D-19-19097R3

Dear Dr. De-Oliveira,

We are pleased to inform you that your manuscript has been judged scientifically suitable for publication and will be formally accepted for publication once it complies with all outstanding technical requirements.

With kind regards,

Jörg Fröbisch, Ph.D.

Academic Editor

PLOS ONE
---

## [Editor Report · Acceptance letter]

18 Mar 2020

PONE-D-19-19097R3 

A New Archosauromorph From South America Provides Insights On The Early Diversification Of Tanystropheids 

Dear Dr. De-Oliveira:

I am pleased to inform you that your manuscript has been deemed suitable for publication in PLOS ONE. Congratulations! Your manuscript is now with our production department. 

With kind regards,

on behalf of

Prof. Jörg Fröbisch 

Academic Editor

PLOS ONE